# Evaluation of Single-Molecule Sequencing Technologies for Structural Variant Detection in Two Swedish Human Genomes

**DOI:** 10.3390/genes11121444

**Published:** 2020-11-30

**Authors:** Nazeefa Fatima, Anna Petri, Ulf Gyllensten, Lars Feuk, Adam Ameur

**Affiliations:** 1Science for Life Laboratory, Department of Immunology, Genetics and Pathology, Uppsala University, 752 36 Uppsala, Sweden; fatima.nazeefa@gmail.com (N.F.); anna.petri@igp.uu.se (A.P.); ulf.gyllensten@igp.uu.se (U.G.); lars.feuk@igp.uu.se (L.F.); 2Department of Epidemiology and Preventive Medicine, Monash University, Melbourne, Clayton, VIC 3800, Australia

**Keywords:** Swedish population, whole-genome sequencing, single-molecule sequencing, nanopore, PacBio, human genome, structural variation

## Abstract

Long-read single molecule sequencing is increasingly used in human genomics research, as it allows to accurately detect large-scale DNA rearrangements such as structural variations (SVs) at high resolution. However, few studies have evaluated the performance of different single molecule sequencing platforms for SV detection in human samples. Here we performed Oxford Nanopore Technologies (ONT) whole-genome sequencing of two Swedish human samples (average 32× coverage) and compared the results to previously generated Pacific Biosciences (PacBio) data for the same individuals (average 66× coverage). Our analysis inferred an average of 17k and 23k SVs from the ONT and PacBio data, respectively, with a majority of them overlapping with an available multi-platform SV dataset. When comparing the SV calls in the two Swedish individuals, we find a higher concordance between ONT and PacBio SVs detected in the same individual as compared to SVs detected by the same technology in different individuals. Downsampling of PacBio reads, performed to obtain similar coverage levels for all datasets, resulted in 17k SVs per individual and improved overlap with the ONT SVs. Our results suggest that ONT and PacBio have a similar performance for SV detection in human whole genome sequencing data, and that both technologies are feasible for population-scale studies.

## 1. Introduction

All human genomes are different, and to a large extent the genetic variation between individuals can be explained as insertions, deletions, duplications, or translocations [1,2]. Taken together, such structural variation (SV) events affect a large portion of the genomic sequence when comparing any given individual to the existing reference genome, and although our knowledge about SVs is still incomplete, they have been shown to contribute to a number of human diseases and conditions [3,4,5,6,7]. In recent years, SV analysis has become a key endeavour for genomics research [8,9,10,11,12] and these efforts will likely lead to an increased understanding of disease etiology and progression and possibly also improved clinical therapy. Although several different methods exist for SV detection in a human genome, whole-genome sequencing (WGS) is at present the most efficient tool to find SVs at base-pair resolution. Large-scale sequencing projects are now starting to provide us with a wealth of information for SVs across different human populations [13,14,15]. However, as of today, most human WGS projects are performed using short-read sequencing, and while this offers sequencing at a low cost per sample, the alignments of short reads to a reference genome can cause ambiguity and pose challenges for SV detection [16,17,18].

Long-read single-molecule sequencing is an alternative approach for SV analysis, which has gained popularity through advances in the ONT and PacBio technologies [19,20,21,22]. With ONT’s high-throughput instrument PromethION, it is feasible to analyse large human sample sets without large upfront investment. Several studies have now been published where the PromethION system has been used for human whole genome sequencing and SV calling [23,24,25,26]. Of these, by far the largest study was performed by Beyter et al. [26], where 1817 individuals from Icelandic population were sequenced on long-read technology, resulting in a median of 23,111 SVs per individual across autosomal chromosomes. PacBio technology has been used for whole genome sequencing since 2015 when the haploid human genome (CHM1) was analyzed using the RS II system [27,28]. Since then, numerous studies used PacBio technology for SV detection in human samples and typically over 20,000 SVs are detected in an individual [29,30,31]. With the latest Sequel II system, which has significantly higher throughput as compared to previous instruments, the per Gigabase (Gb) cost of PacBio continuous long-read (CLR) sequencing has been estimated to 13–26 USD per Gb [32]. This is in a similar range as the estimated 21–42 USD cost per Gb for ONT PromethION. However, it should be noted that these types of cost estimates are challenging since the prices for sequencing reagents, as well as the throughput of the instruments, are constantly changing. Moreover, ONT sequencing can be performed at significantly lower cost for larger projects, due to discounts when ordering multiple flow cells.

The most comprehensive multi-platform SV comparison to date was reported by Chaisson et al. [33], where the performance of seven different sequencing and optical mapping technologies were evaluated in three parent-child trios. The authors recovered over 27,000 SVs per genome, and were able to compile a gold standard dataset of SVs that can be used in the scientific community for comparison and evaluation of SV calls generated in other projects. However, only one of the genomes in the Chaisson et al. study was sequenced with ONT, and no comprehensive comparison between ONT and PacBio SVs was performed. Other than this, to our understanding, there is only one published study where SV calls from ONT and PacBio data has been compared [34]. However, this comparison was based on data from only one individual (HG002).

We previously generated PacBio whole-genome sequencing data to assemble high-quality de novo genomes for one male (Swe1) and one female (Swe2) Swedish individual [29]. In the present study, we performed additional ONT PromethION sequencing of Swe1 and Swe2 and applied a common computational SV detection pipeline both for the PacBio and ONT data. In this way, we could investigate how SV results vary between Swe1 and Swe2, as well as between the two different technologies applied to the same individual.

## 2. Materials and Methods

### 2.1. Sample Collection

The procedure for genomic DNA extraction from whole-blood samples is described in the study by Ameur et al [29]. The blood samples were collected over a decade ago and the two individuals (Swe1 and Swe2) were selected from a group of participants involved in the SweGen project [35]. Sample collection was performed as part of the Northern Sweden Population Health Study (NSPHS), which was approved by the local ethics committee at the University of Uppsala (Regionala Etikprövningsnämnden, Uppsala, 2005:325 and 2016-03-09). In compliance with the Declaration of Helsinki of 1975, study participants provided written informed consent for inclusion to the study including the examination of environmental and genetic causes of diseases.

### 2.2. Whole-Genome Sequencing of Two Swedish Genomes

The DNA libraries were prepared using ONT SQK-LSK109 ligation kit, and sequenced using PromethION flow cell R.9.4.1. Lambda-phage (Accession J02459.1) was used as a control DNA. The sequencing for both individuals was conducted on beta release of PromethION device, and real-time basecalling was performed with Guppy version 1.4.0 (ONT). For each individual, libraries were prepared using both native DNA and DNA sheared to 20 kilobases (kb) using the MegaRuptor system. The native and sheared DNA libraries were run on separate PromethION flowcells. In total, 28 million reads were generated with a yield of 209 Gb. Experiment metrics can be found in Appendix A and Appendix A; visualisation for quality control of ONT data was performed with NanoComp and NanoPlot [36]. PacBio sequencing data was obtained from the 2018 study by Ameur et al. [29].

### 2.3. Alignment and Reference Genome

Genome indexing and alignments were performed with Minimap2 version 2.14-r883 [37]. The human reference genome assembly used for this study is the GRCh38 [38] release (GCA_000001405.15) that does not contain alternative contigs. It represents a non-redundant haploid genome containing a total of 195 sequences; primary sequences of assembled (both autosomal and sex) chromosomes, mitochondrial genome (chrM), (unlocalized) scaffolds with unidentified location in a chromosome, unplaced scaffolds, i.e., sequences with unknown chromosome assignment, and a decoy chromosome of 1718 basepairs (bp) for the Epstein–Barr virus (AJ507799.2).

#### Alignment of ONT and PacBio Reads

The ONT reads were aligned to the human reference GRCh38 using the –map-ont preset that sets a mapping mode suitable for ONT data with k-mer value set to 15. PacBio reads were aligned to the human reference GRCh38 using the –map-pb as preset, that sets a k-mer value of 19 and allows indexing of homopolymer compressed minimsers (k-mers) that essentially means compression of homopolymers to a single base. The alignment statistics were gathered using SAMtools version 1.9 [39] and Qualimap version 2.2.1 [40].

### 2.4. Structural Variant Calling and Analysis

Alignments were analyzed for SVs using Sniffles version 1.0.10 [17]. Sniffles has been tested to perform at a high precision and recall after Minimap2 alignment [25,41]. The minimum SV length was set to 50 bp. Callsets were investigated for overlaps between the SVs detected in ONT and PacBio alignments, and between the SVs detected in two Swedish genomes and high confidence set of SV calls. The high confidence set calls were based on multi-platform high-resolution data generated by Chaisson et al. [33]. For identification of overlaps, SURVIVOR version 1.0.7 [42] was used to merge the SV callsets; maximum distance was set to 500 bp and SV type was enabled. SURVIVOR was also used to collect SV statistics. Data visualisation was performed in R with ggplot2 [43] and VennDiagram [44].

### 2.5. Downsampling Analysis

Downsampling of aligned PacBio reads was performed with Sambamba version 0.7.1 ([45] using the view option; format was set to BAM, fraction of reads was set to 0.5, number of threads was set to 10, and subsampling seed was set to 16. The output files, in BAM format, were used for SV calling that was performed as described in Section 2.4.

## 3. Results

### 3.1. Overview of the Study

The objective of this study was to evaluate the utility of two long-read technologies for detection of germline structural variation in human whole genome sequencing data. For this purpose, we made use of previous PacBio RS II data from one male (Swe1) and one female (Swe2) Swedish individual that we previously sequenced to high coverage [29]. Additionally, we generated ONT data for the Swe1 and Swe2 individuals on the PromethION system. For each individual, one sheared and one unsheared DNA library was constructed and runs were performed on two separate PromethION flow cells. The sheared libraries produced an average 67.54 Gb per flow cell, and an average N50 of 13.1 kb. The unsheared libraries generated less data (36.9 Gb), and read lengths at an average of 23.75 kb. We further merged the data for the sheared and unsheared libraries, generating one ONT dataset for each individual for all further analysis. The overall statistics for the ONT libraries are presented in Appendix A. The read length distributions for the ONT and PacBio datasets are shown in Appendix A.

### 3.2. Mapping of Long-Read Data for two Swedish Individuals

To ensure that the same types of SV events are detected in the ONT and PacBio data, and to reduce potential bias between the datasets, a common analysis pipeline was constructed and applied to all samples (Figure 1). In the first step of this pipeline, Minimap2 [37] was used to align the long reads to the human reference genome GRCh38. The mapping results are shown in Table 1 (detailed alignment statistics can be found in Appendix A). For the PacBio data, we obtained 66.54× and 65.56× average coverage for Swe1 and Swe2, respectively. The ONT data generated lower average coverage; 31.69× for Swe1 and 31.49× for Swe2. Although there are differences in coverage between the two technologies, the coverage reached in the ONT data is above the traditional 30× limit for detection of germline genetic variation from high-throughput sequencing data [46]. Thus, we believe we should be able to identify a large fraction of germline homozygous as well as heterozygous SVs in all of our long-read datasets.

### 3.3. Characterisation of Structural Variation in ONT and PacBio Data

The software Sniffles [17] was used for SV calling from the aligned reads (full SV calling results including parameters are specified in Appendix A). This resulted in a detection of 17,157 (Swe1) and 16,715 (Swe2) SVs of length greater than or equal to 50bp in the ONT data, and 22,829 (Swe1) and 23,284 (Swe2) ≥ 50 bp SVs in the PacBio data (see Table 2) across autosomal and sex chromosomes. This includes a count for translocations and nested events such as inversions flanked by deletions, and inverted duplications that are combinations of main SVs. The length distributions of the SVs are displayed in Figure 2. A peak of insertion and deletion elements was detected around 300 bp, representing Alu repeat elements. Moreover, the majority of SV elements were below 1 kb in length. Both of these observations are consistent with results from previous studies [25,29]. In comparison, the study on the Icelandic population by Beyter et al. [26] detected more SVs per individual in ONT data than what we found in the Swe1 and Swe2 ONT data, a median of 23,111 SVs per individual. This discrepancy can likely be explained by differences in sequencing coverage and analysis parameters between the two studies.

### 3.4. Comparing the Swedish SVs to a High Confidence Set of SVs

To assess the quality of our SVs, we compared our results to the high confidence set of structural variants detected from the multi-platform study by Chaisson et al [33]. In this comparison, the maximum distance between SV breakpoints was set to 500 bp. We found that 15,467 of merged 22,116 ONT SVs (69.9%) overlap with the high confidence set, as compared to 17,418 of 29,800 merged PacBio SVs (58.4%) (Figure 3). The fact that a majority of the structural variations detected in our Swedish genomes overlap with events found in three family (parent-child) trios, which originate from other geographic regions and have been sequenced and analyzed using different methods than our PacBio/ONT data, makes us confident that our analysis pipeline is efficient in detecting true SV events. Moreover, the overlap of SVs between the Swedish genomes and the high confidence set are in agreement with a previous study by Audano et al. [47], where 15,291 SVs events were observed in the majority of the human genomes. It is likely that, as stated in the study, the current human reference genome carries a minor allele or an error at these positions [47].

### 3.5. Comparison of SVs between Sequencing Technologies and Individuals

We further performed a comparison of all the SVs detected in Swe1 (male) and Swe2 (female) to determine which ones are overlapping between individuals and which ones are overlapping between sequencing technologies (Figure 4). When considering insertions, 8814 events were found to be overlapping for Swe1 in both the ONT and PacBio dataset for that individual. The corresponding number for Swe2 was 7785. These numbers can be compared with the insertions that were detected from the same technology but in different individuals: 5388 insertions overlapping between Swe1 and Swe2 for ONT sequencing, and 7963 insertions overlapping for PacBio. Thus, there is a slightly higher overlap between insertions detected in the same individual but with different technologies, as compared to insertions detected with the same technology but in different individuals. When considering deletions, 7162 events were detected for Swe1 both in the ONT and PacBio dataset, and 7165 deletions were detected in Swe2 by both technologies. These numbers can be compared with the deletions that were detected from the same technology but in different individuals: 4699 deletions overlapping between Swe1 and Swe2 when using ONT, and 5617 deletions overlapping with PacBio. This shows that the pattern with a higher overlap for SVs detected in the same individual, as compared to SVs detected in different individuals but using the same technology, is even more pronounced for deletions than for insertions. One possible explanation for this observation is that deletions are easier to correctly call in long-read sequencing data [48]. Overall, the comparisons show that both long-read technologies are good at detecting the same SVs.

### 3.6. Comparison of SVs in ONT and Subsampled PacBio Data

In our study, the PacBio data has about two times higher coverage than what was obtained for ONT. To compare the performance of the two technologies at similar coverage, we performed a random subsampling of the PacBio data to obtain ~33× coverage for each of Swe1 and Swe2, which is similar to the coverage in the ONT datasets. Unsurprisingly, fewer SVs were called from the downsampled PacBio data (17,289 for Swe1 and 16,848 for Swe2) as compared to from the complete PacBio data for Swe1 and Swe2. The SV events from the downsampled PacBio data displayed a high overlap with the ONT SVs, both for insertions (77–78%) and for deletions (85–86%) (Figure 4B). These numbers can be compared to the corresponding SV overlaps when using all the PacBio reads, which are in a range of 67–69% for insertions and 81–82% for deletions. Our conclusions from this analysis are that ONT and PacBio detects a similar amount of SVs when sequencing is performed at around 30× coverage, and that SVs found in the downsampled PacBio data are in good agreement with the ONT SV calls.

## 4. Discussion

This study reports findings from the first PromethION sequencing runs of Swedish human genomes, as well as the analysis of SVs in the same human individuals sequenced on two different long-read platforms. One limitation of this study is that the PacBio sequencing was performed on an older generation instrument (RSII). Possibly, the PacBio results would be somewhat different on the Sequel I or II instruments, and in particular if the recent protocol for highly-accurate long-read HiFi sequencing was used [49]. However, it should be noted that we obtained a high coverage in the RSII PacBio data (>65×), sufficient even for creating high quality de novo genome assemblies [29]. For this reason, and since the underlying technology is the same in the RSII and Sequel systems, we believe that the RSII dataset can be used to represent PacBio’s CLRs. Regarding the ONT data, it was generated on the state-of-the art PromethION instrument but at a coverage lower than that for PacBio and this could possibly explain why fewer SVs were detected in the ONT data. Recent updates of the ONT basecalling software could have increased the quality of the reads and resulted in improved SV results. Despite these limitations, which highlight the difficulty to perform cross-platform comparisons in this rapidly developing field, we believe our PacBio and ONT datasets are of a sufficient quality and coverage depth to jointly analyse and compare SVs detected from the two technologies.

Another challenging aspect of this study was to compare SVs between samples and determine whether two partially overlapping SVs should be considered as the same event. We relied on the software SURVIVOR [42] for the SV comparison, and allowed the start and end coordinates to differ with at most 500 bp. This is an important parameter to determine breakpoints and detect overlaps between SVs. We performed testing for maximum allowed distance with multiple values and noticed a trade-off between the number of merged and overlapping SV events across samples Appendix A). Since the aim of our study was to find broad trends between different technologies, rather than to examine individual SV events, the uncertainty about which SVs are overlapping does not have a major impact on our results and conclusions. For this same reason, we chose to only use the SV caller Sniffles [17] which has been shown to perform well on long-read data [25], instead of also using other SV calling methods [30,34,41,50] that would likely give slightly different results. However, for projects aiming to determine all SVs occurring in a population, or to understand the functional consequences of SVs and their relation to medical conditions or diseases, it will be crucial with a robust and standardised method for identification and comparison of SVs between different individuals and samples.

As long-read sequencing technologies continue to improve and become more accessible, we envisage that large human structural variation databases will be constructed for various populations and cohorts, similar to what is currently available for short-read sequencing [9,13]. Already now, a few population-scale initiatives have been initiated but the number of such projects is likely to grow in the coming decade. In a future where SV calling from long-read data becomes a standard approach to examine human genetic variation, it will be important to evaluate the available technologies and determine which strategy is currently the best option for detection of different types of SVs. Our results suggest that both PacBio and ONT are viable options for SV calling in human germline samples.

## Figures and Tables

**Figure 1 genes-11-01444-f001:**
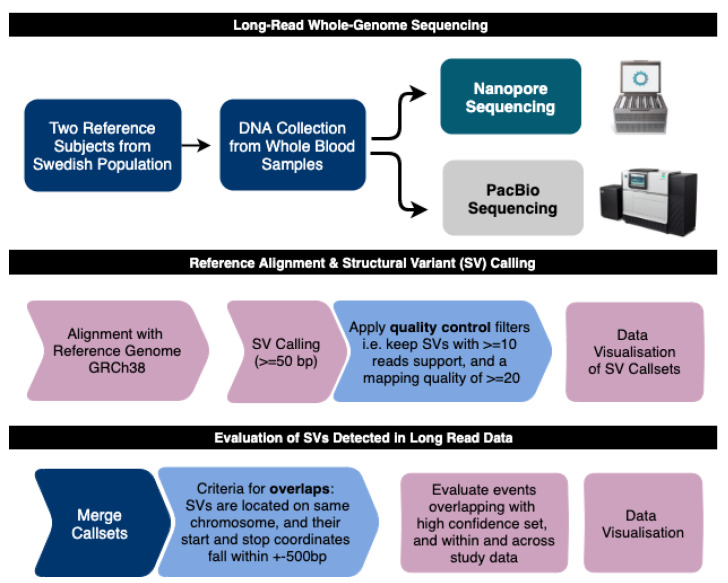
Study Overview: Long-Read Sequencing of Swedish Genomes. Analysis steps for detection and evaluation of structural variants in the two Swedish human genomes (Swe1 and Swe2).

**Figure 2 genes-11-01444-f002:**
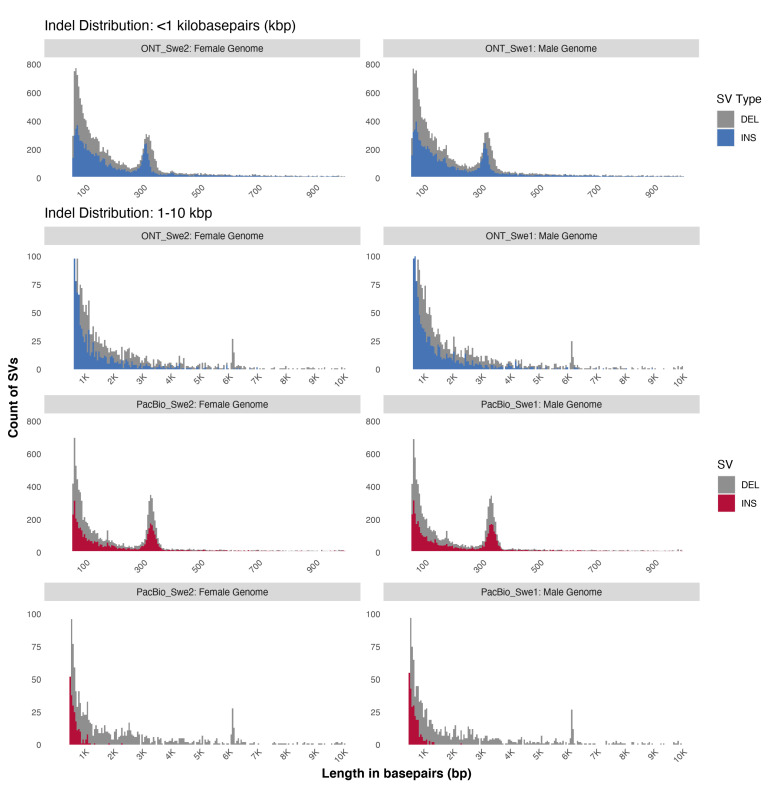
Length profiles of SVs identified in Swe1 and Swe2. The four panels at the top show length distribution for SV insertions and deletions detected in the ONT data, up to 1 kb on the first row and 1–10 kb on the second row. The four panels at the bottom show corresponding SV length distributions in the PacBio data. The plots were generated with custom R script (available on GitHub at github.com/Nazeeefa/NanoSwe).

**Figure 3 genes-11-01444-f003:**
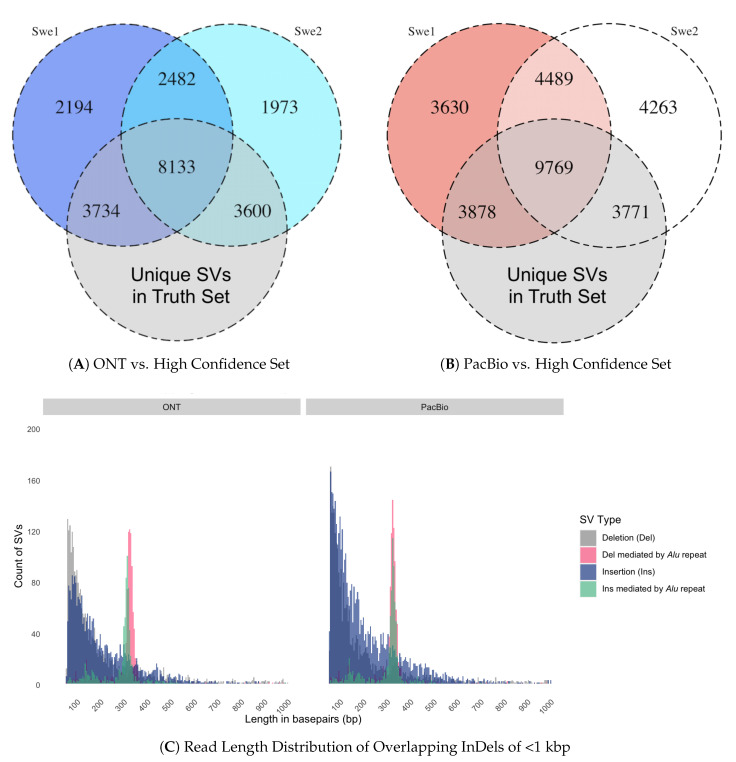
Comparison of Swedish SVs to a high confidence set of SVs. (**A**) Venn diagram showing the overlap of Swe1 and Swe2 SVs detected by ONT data with a high confidence set of SVs from the study by Chaisson et al, 2019. (**B**) Overlap of SVs detected in PacBio Swe1 and Swe2 datasets with the high confidence set. The figures were generated with the R package VennDiagram [44]. (**C**) Length distributions of the SVs, in Swe1 and Swe2, which are overlapping with the high confidence set. The plots display the SVs (only indels and associated Alu repeat element) that overlap with the high confidence set for ONT data (left) and PacBio data (right). The scale on the x-axis represents length of overlapping SVs in a range 50 bp–1 kb.

**Figure 4 genes-11-01444-f004:**
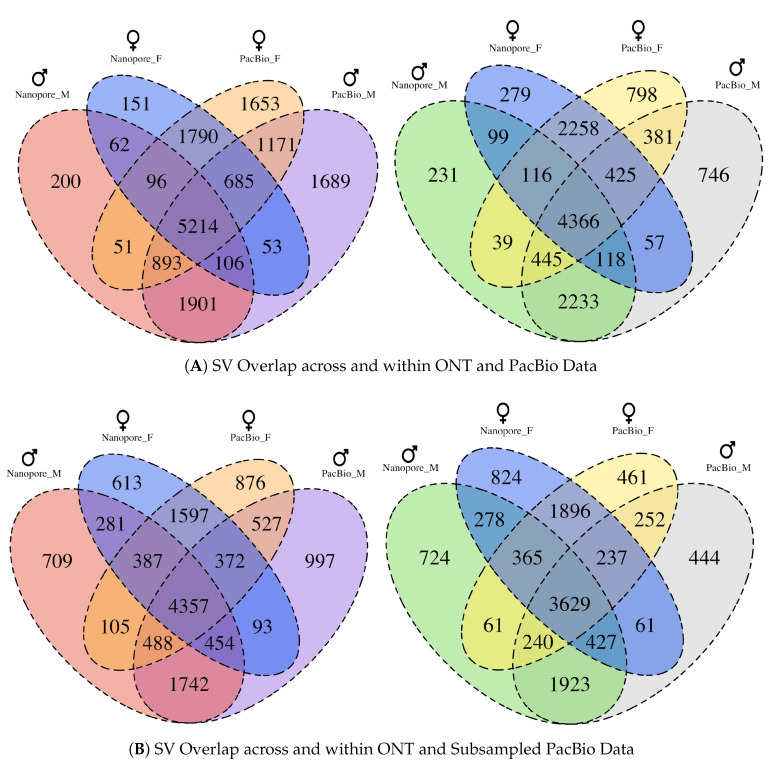
Comparison of SV events detected in Swe1 and Swe2 using ONT and PacBio sequencing. (**A**) Insertions (left) and deletions (right) detected within and between the Swe1 and Swe2 individuals from the two long-read technologies (ONT and PacBio). (**B**) Same comparisons for insertions (left) and deletions (right) as (**A**), but involves the PacBio data subsampled to 33.28× coverage for Swe1 and 32.78× for Swe2. The figures were generated with the R package VennDiagram [44].

**Table 1 genes-11-01444-t001:** Alignment statistics for the long-read datasets for Swe1 and Swe2.

Sample	Total Sequence Data, Gb	Mean Mapping Coverage	GC Content, %
ONT_Swe1	104.5	31.69×	40.72
ONT_Swe2	104.4	31.49×	40.41
PacBio_Swe1	235.7	66.54×	40.41
PacBio_Swe2	232.9	65.56×	40.46

**Table 2 genes-11-01444-t002:** Summary of SVs detected in Swe1 and Swe2.

SV Type	ONT Swe1	ONT Swe2	PacBio Swe1	PacBio Swe2
Deletions	7769	7820	9052	9101
Insertions	8746	8369	12,441	12,218
Duplications	185	147	331	291
Inversions	133	128	196	193

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
