# Peer review of "Evaluation of Single-Molecule Sequencing Technologies for Structural Variant Detection in Two Swedish Human Genomes"

_genes, 2020, doi:10.3390/genes11121444_

Round 1
Reviewer 1 Report
Evaluation of Single-Molecule Sequencing Technologies for Structural Variant Detection in Two Swedish Human Genomes
The authors of the paper performed a comparison between different technologies to identify SVs (ONT and PacBio CLR), they used manimap2 for aligning and sniffles plus survivor to identify and compare SVs between the samples. The paper showed that both technologies have similar performance in identifying SVs, the introduction and methods are written in an easy way to follow, but I have the following concerns:
- The coverage of Pacbio is ~2 folds higher than ONT, Should it be downsampled and see how that will affect the results?
- It will be also informative if the authors make a comparison to show the effect of read-coverage on SV detection, as the paper main message is to show how different long-read technology performs on identifying SVs
- The authors did not mention the tools they used for ONT base-calling and which version (the updates in base-calling tools has a great effect on the yield)
- What was the read length distribution in each tech, as the authors mentioned the large read length of the technologies makes it suitable for SV detection
- The authors dependent only on one SV caller for their analysis (Sniffles), while using different tools, ex: SVIM or/and NanoSV could make a more comprehensive comparison.
Author Response
Please see the attachment for our point-by-point response to the reviewer's comments

Reviewer 2 Report
The authors perform an evaluation of how long-read sequencing data from either Pacific Biosciences and Oxford Nanopore Technologies can be used for SV detection. Both platforms are increasingly used on larger populations, so such a comparison, if executed properly, would be informative.
The authors identified a large difference in the number of SVs (ONT: 17k vs PB: 24k), which in my opinion does not indicate that both technologies perform similarly for SV detection as the authors conclude.
Please find my detailed comments below.
Major comments
- The PacBio dataset has double the coverage of the ONT data in your comparison, which definitely biases the downstream analysis. The coverage is substantially different, but you apply the same required number of reads supporting an SV in Sniffles. I would recommend randomly downsampling the PacBio reads to match the coverage of the ONT data (e.g. with samtools).
- How do the authors explain that their Swedish genomes share so many variants with the dataset from Chaisson et al? Are all these variants highly frequent and present across populations? Or is the maximum distance between breakpoints of 500 bp not strict enough? Does this comparison take the variant type into account? I think it's important to be more specific with regard to which parameters were used for SURVIVOR. Alternatively, you could try Jasmine which also takes the inserted sequence into account (https://github.com/mkirsche/Jasmine). I would not expect that 70% of your variants can be found in other populations - this goes against the observation from e.g. Audano et al that most structural variation is rare. Are your genomes equally similar to the three individuals from Chaisson, or more closely resembling one of them?
Minor comments
- The ONT basecaller Guppy changes constantly, it would be important to mention which version was used for basecalling the data.
- The introduction mentions that with the increased throughput of the Sequel II the costs of PacBio and ONT are roughly the same. Perhaps my numbers are outdated, but especially when buying ONT flow cells in bulk I believe there is still a substantial price difference. It would probably be informative to be more specific here with regard to costs.
- Please make sure to cite all tools used in your study. I don't want to be that reviewer unnecessarily asking for a citation of one of my papers, but the citation for NanoPlot/NanoComp/etc is missing, and given the supplementary data you clearly used these tools. I also noticed that the surnames are incorrect in reference 25, but are correct in reference 24.
Author Response

(The authors gave the same response as above.)

Round 2
Reviewer 2 Report
I wish to congratulate the authors on their improved manuscript, and I am largely satisfied with the improvements. While the version of the guppy basecaller is severely outdated I am not going to ask to repeat the entire analysis with a more current version, although I do think it would be informative to see the impact on SV calling and also make this publication more representative of the current state of the technology.
As expected the lower coverage did also reduce the number of structural variants. Therefore one issue remains, as in the current version, the abstract is misleading, and should either explicitly state that the comparison is at different coverage levels, or better, including the numbers from the downsampled analysis to clearly represent the findings.
Author Response
Thank you for the positive comments. We agree that the coverage differences between ONT and PacBio, as well as the downsampling results, are of importance. We have now included the coverage levels for ONT and PacBio in the abstract and also added the following sentence in the abstract: “Downsampling of PacBio reads, performed to obtain similar coverage levels for all datasets, did not have a major impact on the SV results.”